# Spatio-temporal based deep learning for rapid detection and identification of bacterial colonies through lens-free microscopy time-lapses

Paul Paquin[1]*, Claire Durmort[2], Caroline Paulus[1], Thierry Vernet[2], Pierre R. Marcoux[1], Sophie Morales[1]

**1** Univ. Grenoble Alpes, CEA, LETI, DTBS, LSIV, 38054 Grenoble, France, **2** Univ. Grenoble Alpes, CNRS, CEA, IBS, 38054 Grenoble, France

* paul.paquin@cea.fr

**Data Availability Statement:** The minimal dataset underlying this study is available at https://github.com/paulpaq98/Holographic_Database.

## Abstract

Detection and identification of pathogenic bacteria isolated from biological samples (blood, urine, sputum, *etc.*) are crucial steps in accelerated clinical diagnosis. However, accurate and rapid identification remain difficult to achieve due to the challenge of having to analyse complex and large samples. Current solutions (mass spectrometry, automated biochemical testing, *etc.*) propose a trade-off between time and accuracy, achieving satisfactory results at the expense of time-consuming processes, which can also be intrusive, destructive and costly. Moreover, those techniques tend to require an overnight subculture on solid agar medium delaying bacteria identification by 12–48 hours, thus preventing rapid prescription of appropriate treatment as it hinders antibiotic susceptibility testing. In this study, lens-free imaging is presented as a possible solution to achieve a quick and accurate wide range, non-destructive, label-free pathogenic bacteria detection and identification in real-time using micro colonies (10–500 μm) kinetic growth pattern combined with a two-stage deep learning architecture. Bacterial colonies growth time-lapses were acquired thanks to a live-cell lens-free imaging system and a thin-layer agar media made of 20 μl BHI (Brain Heart Infusion) to train our deep learning networks. Our architecture proposal achieved interesting results on a dataset constituted of seven different pathogenic bacteria—*Staphylococcus aureus (S. aureus)*, *Enterococcus faecium (E. faecium)*, *Enterococcus faecalis (E. faecalis)*, *Staphylococcus epidermidis (S. epidermidis)*, *Streptococcus pneumoniae R6 (S. pneumoniae)*, *Streptococcus pyogenes (S. pyogenes)*, *Lactococcus Lactis (L. Lactis)*. At T = 8h, our detection network reached an average 96.0% detection rate while our classification network precision and sensitivity averaged around 93.1% and 94.0% respectively, both were tested on 1908 colonies. Our classification network even obtained a perfect score for *E. faecalis (60 colonies)* and very high score *for S. epidermidis at 99.7% (647 colonies)*. Our method achieved those results thanks to a novel technique coupling convolutional and recurrent neural networks together to extract spatio-temporal patterns from unreconstructed lens-free microscopy time-lapses.

**Funding:** The authors received no specific funding for this work.

**Competing interests:** The authors have declared that no competing interests exist.

## Author summary

In order to combat the spread of resistances and to meet the specific needs of infection diagnosis in rural and remote settings, there is a strong need for new diagnostic tools that are faster, space efficient and less expensive than conventional ones while still being accurate. To answer those criteria, optical identification methods such as lens-free microscopy seem best-suited. Indeed, lens-free microscopy is an imaging technique that allows for continuous image acquisition during incubation, as it is non-destructive and compact enough to be integrated into an incubator. Here, we propose the combined use of lens-free microscopy imaging and deep learning to train neural networks able to detect and identify pathogenic species thanks to bacteria growth time-lapses on thin-layer agar and a novel deep learning architecture combining recurrent and convolutional neural networks. We achieved high quality detection and identification from 8 hours onwards on a dataset composed of 7 different pathogenic strains. Our study provides new results proving the usefulness of lens-free imaging on pathogenic species for rapid detection and identification tasks in low-resource settings.

## 3. Introduction

The ever-increasing number of multidrug-resistant bacteria is having a growing influence on mortality worldwide. With the emergence of resistance to Colistin, a last resort antibiotic effective against Gram-negative bacteria and especially the *Enterobacterales*, at the end of 2015 [1,2], the situation is worsening. Unnecessary usage of wide-spectrum drugs is accelerating the global spread of multidrug resistant organisms. Antibioresistance is even more alarming when it comes to last-resort antibiotics. Infections resulting from those resistant bacteria are more and more difficult to cure. Such is the case with *Escherichia coli*, which is seeing Colistin resistances emerge in already multidrug-resistant strains [3]. The World Health Organization predicts that by 2050 infectious diseases will once again be the leading cause of death in the world with 10 million deaths annually [4].

Rapid identification of isolated pathogenic bacteria in biological samples (urine, blood, sputum, etc.) is a key step in the diagnosis of an infection, and therefore in its management. Indeed, speed and reliability of identification determine how quickly a specific and optimised antibiotic therapy–more effective and less likely to generate new resistance to antibiotics–will be administered by precisely targeting the identified pathogen.

Current solutions for microbial species identification providing high quality outputs still have many shortcomings resulting from many concessions done in order to achieve those very high identification performance. That is to say, they often require complex, time-consuming and costly identification protocols in exchange for accurate results.

Identification methods commonly used are either based on biochemical profiling by using identification cards such as Vitek 2 [5] or on time-of-flight (TOF) mass-spectrogram in particular with matrix assisted laser desorption ionization (MALDI-TOF) [6,7]. Those techniques usually require isolated colonies grown on overnight cultures; this requires users to wait 18 to 48 hours for at least one identifiable colony to fully incubate. There is also genotypic analyses, which allow identification of a target bacteria or virus thanks to its genomic sequence: the most popular methods using this principle are the sequencing of 16S [8] for prokaryotes and 18S ribosomal Ribonucleic acid (rRNA) [9] for eukaryotes or PCR-based methods for rapid identification [10]. Such analyses allow reliable and robust pathogenic bacteria identification

in samples; nevertheless, it is at the expense of an *a priori* knowledge and understanding of the infection.

For this reason, the current lines of research on microbial identification are diversified. The first direction is the development of identification techniques that address the specific needs of low-resource settings (LRS) or small isolated laboratories [11]. These techniques would expand the use of diagnosis prior to antibiotic therapy, thereby reducing unnecessary use or misuse of antibiotics. The second strand of research focuses on automation: many steps in the entire process have not yet been automated and require human processing. For instance, in the case of MALDI-TOF, the choice of the colonies to be analysed, their sampling, the mixing with specific reagents, are several tricky steps to automate. A fully automated identification method would lead to a "smart incubator", [12] *i.e.* an incubator able to provide an automatic identification on every incubated plate, as soon as enough phenotypic information is available. In that manner, identification results could be delivered during plate incubation, usually at night, and all human processing done the day after could be dedicated to antibiotic susceptibility testing (AST). The third and last area of research is single-cell characterization [13–16]: if a technique can provide an identification on very low biomasses, typically 100 cfu (colony-forming unit) and less, the results could be obtained before the incubation step, which would reduce time-to-result significantly (18 h to 48 h less).

Optical identification methods, particularly when they do not require labelling techniques (i.e. fluorescence microscopy), have decisive advantages: non-invasive, rapid and non-destructive, they could potentially allow automated, reliable, low-cost and high-throughput diagnosis. These techniques are based on various optical properties such as scattering, absorption and emission, reflection or even phase contrast each of which allows for the study of numerous bacterial characteristics [17,18] (S1 Table). On the one hand, spectroscopic methods highlight bacterial biochemistry by collecting spectral data over a certain range through the use of absorption and emission (Fourier transform infrared spectroscopy) [19] or inelastic scattering (Raman microspectrocopy [20,21]). On the other hand, scattering methods tend to focus on colony morphology due to their use of diffraction and reflection principles to produce spatial data. In particular, elastic scattering [22,23] and more precisely forward scattering (BARDOT) [24,25] both provide specific direct two-dimensional patterns for each bacterial colony based on their shape, which makes these techniques sensitive to intra-species variation. Furthermore, some optical setups are capable of producing both spectral and spatial data thanks to hyperspectral approaches over large field of view, some even imaging full standard petri dishes. For instance, hyper spectral diffuse reflectance microscopy [26,27] allows the entire dish to be captured and its spectral signature to be acquired for later use thanks to in line scanning. Nevertheless, these systems must wait for colonies to reach a certain size for them to be imaged and are not compact enough to be placed inside an incubator to obtain real-time images. This leads to a delay in the detection and identification processes.

Lens-free imaging allows for wide-range detection and identification through direct diffractograms analysis [28] or holographic reconstructions [29,30]. Thanks to "colony fingerprint" (*i.e.* discriminative parameters) and through cluster analysis, Maeda *et al.* [31] showed that discriminating microorganisms was possible with lens-free imaging within the visible range after 8h of incubation on a limited dataset. In the same manner, Wang *et al.* [32] proposed an automated lens-free imaging system to detect and classify bacteria by using neural networks on a small time-lapse (2 h) obtaining impressive results (>95% detection rate and 99.2–100% precision within 12h) on 3 different *Enterobacterales* (*Klebsiella pneumoniae*, *Escherichia coli*, *Klebsiella aerogenes*).

In this paper, micro colonies long kinetic growth patterns (>6 h) are used to train a two-stage deep learning architecture that will detect and identify pathogenic bacteria over a wide-field, through growth time-lapse, in less than 12 hours.

This method differs from other lens-free techniques by its approach based on thin-layer un-reconstructed video microscopy for detection and classification on some of the most prevalent anaerobic pathogenic bacteria in clinical settings. Among other things, it differentiates itself through its use of a thin-layer culture medium that minimises the sample-sensor distance. This technique works well in a low-density context (at maximum ~ 200 colonies/mm$^2$) as it is possible to image the sample over a large field-of-view while avoiding a magnification-induced superimposition of holograms which occurs when the sample is moved away from the sensor. This gives access to a better resolution and higher frequency information than on a Petri dish during a single-shot acquisition. Nevertheless, this technique's sample preparation protocol includes a pour-plating step, thus colonies do not necessarily grow on the same plane as there is a height variability.

This is where our un-reconstructed holographic approach shines by avoiding either making an a priori estimation of the sample-sensor distance or having to produce multiple reconstructions at different heights. It also removes the need for auto-focus, which is not the case for Petri dishes monitored over time due to inter-plate (Agar pouring height) and intra-plate height variability (Agar drying out in the incubator) during an experiment. This method thus avoids all the problems inherent with holographic reconstruction (phase wrapping, twin images, etc.) by learning directly from a series of holograms within a height range. Finally, it is the use of neural networks throughout the processing pipeline that allows this method to obtain rapid and robust results, on par with other work. This pipeline consists of a region-based convolutional network for detection and a novel dual association of a spatial encoder (3D convolutional network) with a temporal encoder (2D recurrent network).

## 4. Materials and methods

### 4.1 Studied species

In this study, seven different bacterial strains were used in the training and testing dataset: Staphylococcus aureus (S. aureus), Enterococcus faecium (E. faecium), Enterococcus faecalis (E. faecalis), Staphylococcus epidermidis (S. epidermidis), Streptococcus pneumoniae R6 (S. pneumoniae), Streptococcus pyogenes (S. pyogenes), Lactococcus Lactis (L. Lactis) (Fig 1). Our S. pneumoniae R6 and E. faecalis OG1RF strains are both ATCC strains, respectively ATCC BAA-255 and ATCC 4707, while other strains are derived from clinical isolates. In particular, our E. faecium D344 strain is derived from the same clinical isolate used in one paper written by Williamson R, et al [33].

### 4.2 Setup

**4.2.1 Material and geometry.** A six wells lens-free microscope, the cytonote6W (Iprasense, France), was used to perform image acquisition. This commercial imaging system can easily fit inside an incubator and record the bacterial growth process thanks to its relatively small size (16x12x10 cm).

It uses a multichip light emitting diode (LED) as its light source, delivering a multi-wavelength partially coherent illumination. During this study, the red wavelength provided, centred on 636 nm with a 25 nm spectral bandwidth, will be used. This light source is located more than 5 cm away from the studied sample and above a 150 μm pinhole placed just under in order to spatially filter the light, scatter the light and obtain a semi-coherent source. This imaging system uses a CMOS (Complementary metal-oxide-semiconductor) image sensor, with a 29.4 mm$^2$ (6.41×4.59 mm) field of view (FOV), 3840×2748 pixels per image and a pixel size of 1.67 μm.

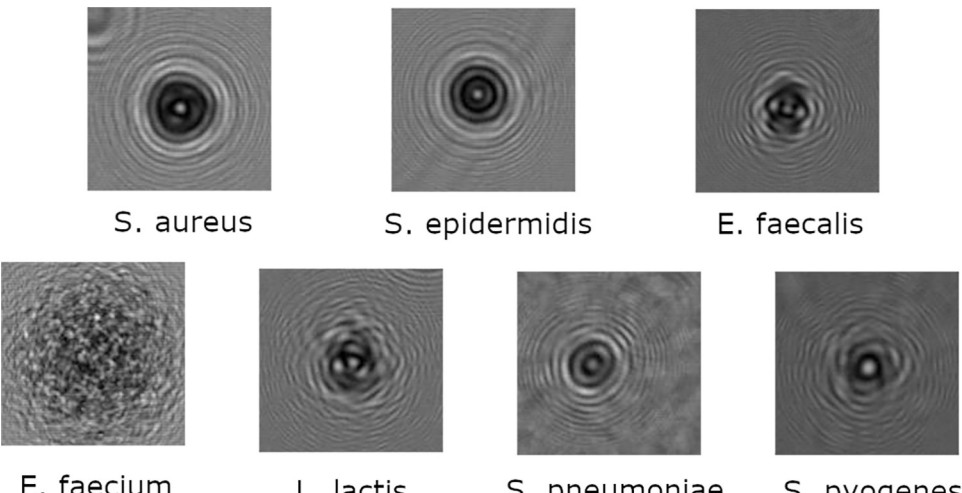

**Fig 1. Bacteria Holograms.** Holographic pattern of 7 bacteria strains after 12 hours of incubation at 37°C on Brain Heart Infusion (BHI) agar.

Bacterial cultures are then put on top of the CMOS, which means micro-colonies are the nearest they could be from the sensor with a distance between 0 and 5 mm depending on the recipient used which might be petri dishes, multiwells plates or in this case thin-layer cultures (~ 150 μm). Thin-layer cultures are 19 μl gas-tight glass frames, which permit anaerobic growth of bacteria inside the chamber containing the culture medium for micro-colonies.

**4.2.2 Lens-free microscopy.** This equipment and geometry let us use lens-free imaging techniques to follow living cells for detection and identification purposes without being intrusive or destructive. This minimalist method consists of acquiring with an imaging sensor the diffraction patterns formed by the interference of the light scattered by the sample and the light passing directly through the sample. This way, a holographic pattern is obtained; it is not an actual image showing the shape of cells but it allows for many applications (counting, tracking, identification, etc.) [30–32,34,35]. A lens-free microscope acquires the hologram intensity of the studied sample which can then be used to reconstruct the original image thanks to some computational algorithms [36,37]. In this study, unreconstructed holograms amplitudes will be shown to be more than enough for accurate detection and identification.

**4.2.3 Biological protocol.** Every bacterial culture was handled according to safety rules specific to a Biosafety level 2 laboratory. Bacteria were grown in brain heart infusion broth (BHI medium) from glycerol stocks stored at -80°C until the optical density (OD) at 600 nm equals 0.3. Each bacterial culture was incubated at 37°C with or without shaking in a 5% $CO_2$ atmosphere depending on species requirements.

Agarose pads were made with BHI medium containing 13 mg/ml low gelling temperature agarose suitable for cell culture (Sigma-Aldrich, A4018). BHI agarose was then liquefied at 75°C and left to cool. When the temperature reached 39°C bacteria were added at a concentration of $2\times10^4$ cfu/ml. As shown in Fig 2, a 19 μL suspension is placed on a 24×24 mm glass slide in a 1 $cm^2$ frame with a 150 μm thickness then another glass slide is placed, this time 22×22 mm on top without trapping bubbles between the slides.

### 4.3 Data acquisition

Following the previously described protocol, a database of holographic time-lapses from mono-microbial cultures was collected by monitoring the growth of micro-colonies in several

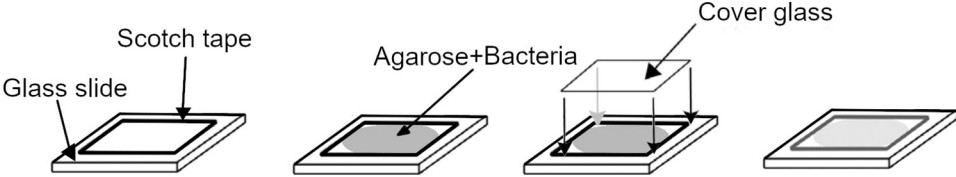

**Fig 2. Thin-layer cultures.** Step-by-step production of suitable thin-layer cultures with agarose pads and BHI medium.

thin-layer cultures. To observe a culture growth kinetic with a Cytonote an image is recorded every thirty minutes. Those experiments lasting twelve hours, each time-lapse is thus composed of 25 images. This database consists therefore of 26 experiments gathering a total of 650 potential images to learn on.

However, each preculture does not show the same biomass (cfu/mL) when inoculated into the BHI agarose medium. The number of observable bacterial colonies on the reduced field of view (29.4 mm$^2$) will therefore be culture-dependant. Thus, for the same turbidity (with an optical density of 1), each species will have a cfu/ml ranging from $2\times10^8$ to $14\times10^8$ cfu/mL, *i.e.* a number of colonies varying from 50 per experiment (*e.g. E. faecium*) to 300 per sample (*e.g. S. aureus, S. epidermidis*). Moreover, the number of experiments per strain is not equivalent, for example, 5 experiments were done with *S. epidermidis* or *E. faecalis* while only 4 were done with *S. aureus*.

In the end, those 39 experiments translate to 5961 micro-colonies: 4053 for training and validation (26 experiments) and 1908 for testing purposes (13 experiments).

### 4.4 Image pre-processing

Images acquired by Iprasense imaging system are quite large (3840×2748 pixels); therefore, every image was cut into forty-nine 960×867 pixels overlayed sub-images (S1 Fig) after confirming that no species-dependent macro-phenomenon occurred. This overlaid subdivision is comprised of an initial 4x4 matrix to which we add 3x4 (half step offset along the x-axis), 4x3 (half step offset along the y-axis) and 3x3 (half step offset in both x-axis and y-axis) matrixes. Raw data was also modified; pixel intensity was changed as pixel values were normalized between a 0–255 range using a min-max normalization technique (also called feature scaling) to carry out a linear transformation from the original data to a chosen scale.

For the detection phase, the task of the neural network was to locate areas of interest and eventually to determine their shapes. Considering this, datasets needed to contain position and shapes of micro-colonies to be used as ground truths. For that matter, images were annotated following Microsoft COCO format (Common Objects in COntext) [38] one of the most popular data format for annotating objects. It is widely use because it is adapted for both object detection and segmentation tasks and due to the Microsoft COCO dataset which is a benchmark for evaluating computer vision models performance. Thus, data annotated in this manner can be repurposed for further trainings of state-of-the-art neural networks to compare results for instance. In this format, objects shapes are described as lists of points forming masks contained in bounding boxes. Colonies masks were done manually, by circling around holographic patterns after a 12-hour incubation and back-propagating them to earlier frames.

### 4.5 Deep learning architecture

Python 3.7.0 [39] was chosen to implement each network alongside with Tensorflow 2.1.0 [40] and the 2.3.1 Keras interface [41]. By following the proposed architecture in Fig 3, the problem

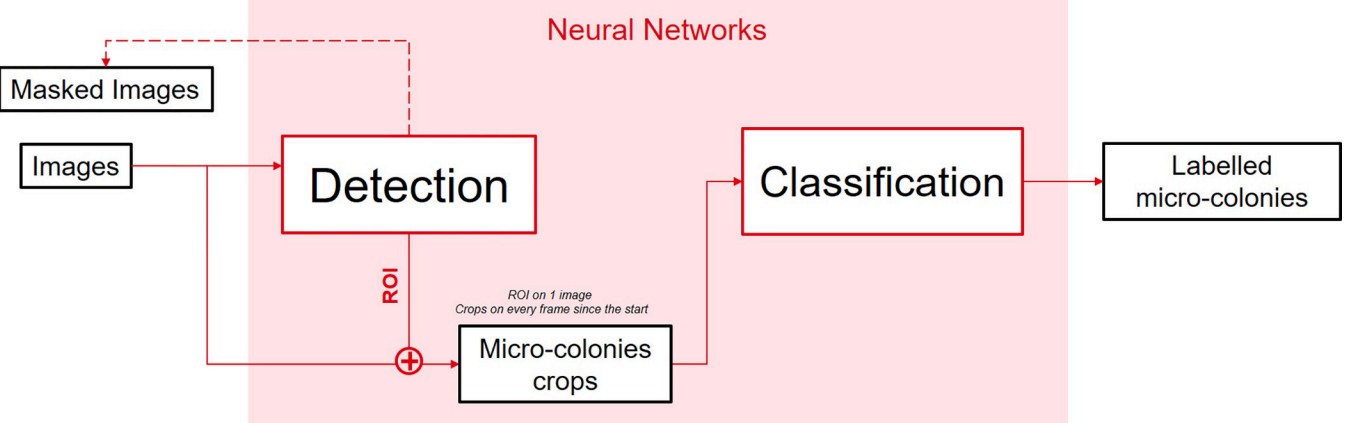

**Fig 3. Global Deep Learning schematized architecture.** Detection and classification are separated into two different tasks. The detection network takes 768×768 pixels sub-images and outputs regions of interest. The classification network uses candidate crop time-lapses as inputs and outputs a probability vector to label each time-lapse.

was decomposed into two distinct tasks to facilitate and to better monitor trainings. Trainings were done with a Nvidia Titan XP as GPU (*graphics processing unit*) using CUDA (Compute Unified Device Architecture) 11.2 [42]. Every training done had an 80/20 repartition between training and validation data.

**4.5.1 Detection neural network.**   The first task identified is to detect growing micro-colonies while they are still incubating. For that reason, a neural network capable of locating small areas of interest on much bigger images is needed. This network must be capable of taking an image as an input and produce multiple lists of points that will then be used to crop bounding boxes for the classification network (S1 Fig).

As described earlier, each image is cut in a 7×7 overlapping sub-images matrix to enable quicker and less memory-consuming inferences. Due to this sub-devising technique many detection inferences give redundant outputs–some colonies are detected multiple times and thus many bounding boxes overlap each other. To answer this problem, a non-max suppression technique was implemented to remove lower quality redundant predictions. This filter consists of taking the predictions with the highest confidence score and comparing their IOU (intersection over union) with other predictions. If the IOU is higher than a certain threshold (0.3 in this case), the compared image is then removed from the prediction list in order to retrieve only different and high-quality predictions.

For the training step, 26 experiments out of the seven selected strains were selected. In particular, the training was focused on a sub-dataset of images between the 11[th] and 21[th] frames, which means between 5 and 10 hours of incubation. Each image was decomposed into 49 sub-images (960×867 pixels) and only ten frames by experiment were utilized, thus the training dataset consisted of 12 740 images to train from. In the same manner, 13 experiments were used for the testing phase, which is equivalent to 6860 testing images.

For the detection neural network, a Mask-RCNN (Mask Regional Convolutional Neural Network) [43] was chosen to propose regions of interest for the next network to work on. For its convolutional model we had recourse to a ResNet-101 (Residual Network) [44], pre-trained on ImageNet [45] and re-trained on our own dataset. To train this network each 960x867 image was resized to be 768x768.

Our Mask-RCNN was trained for 75 epochs on each layer, with a learning rate scheduler that reduced the learning rate on each loss function plateau and an additional early stopping if

the plateau lasted too long to prevent overfitting. The training started with a learning rate of 0.01, this ratio was then multiplied by 0.1 each time the loss function—applied to the validation set—did not fluctuate by more than 0.001 over 10 epochs. Futhermore, if it lasted more than 15 epochs without significant improvement despite the learning rate variation then the training was aborted via early stopping.

**4.5.2 Classification neural network.** The detection neural network outputs a bounding box list when presented with an image. From this list, time-lapses are created by extracting the corresponding crops from each frame available until now. Colony crops are made by retrieving the centre of the bounding box proposed by the detection network and by cropping a 64×64 square around that centre. The new input is thus a holographic time-lapse that varies in length depending on which image the first network was applied on.

From those time-lapses, the classification network must distinguish different species, which implies having a label for every time-lapse as an output. The classification network needs to be able to extract spatial and temporal patterns to identify every species in the database. Considering this goal, a combination of a three-dimensional convolutional [46,47] and a two-dimensional long short-term memory network (LSTM) [48] were chosen, in a similar fashion to a ConvLSTM model [49] which would be applied to an image. This network extracts spatial features from frames with its convolutional part (Conv3D) and labels the whole sequence thanks to its recurrent part which extracts temporal features (LSTM) [50].

One major problem is that this architecture is very size-dependant and kernel size cannot be the same for every network. That is why the structure of the classification network should be dynamic depending on time-lapse length. The basic structure of this model is a pair of Conv3D and ConvLstm2d layers followed by a flatten layer and two dense layers so the output is a probabilistic distribution (Fig 4). This list size is equivalent to the number of classes in the

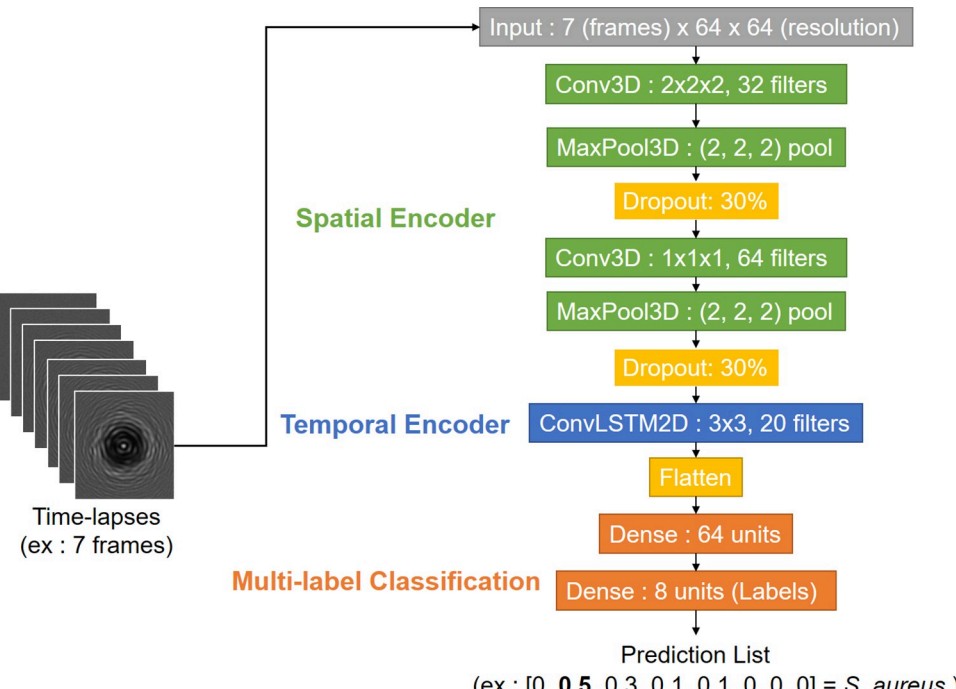

**Fig 4. Architecture of a seven-frame classification network based on a Conv3D+LSTM2D principle.** The network is composed of three parts: a spatial encoding phase with convolutional layers, a temporal encoding phase with a recurrent layer and dense layers to classify the data.

training dataset (seven strain classes and one background class), its values are in percentages (from 0 to 1), its sum equals 1 and each value corresponds to how probable the time-lapse belongs to the class. Some extra layers might be added in some cases; if the input size allows it, a MaxPooling layer is added each time a convolutional layer is used and a dropout layer is also added after each pooling.

For smaller time-lapses (0-1h30) only one Conv3D layer is possible: composed of 32 neurons and a (1, 1, 1) kernel, it is associated to a 20-filter ConvLSTM2D with a (3,3) kernel. For bigger time-lapses (>7h30) larger (3, 3, 3) kernels and more neurons (64 at most) can be utilized. It is possible to add pooling layers with (2,2,2) kernels and to put three (Conv3D + Max Pooling) layers in the spatial encoder too. Bigger 40-filter ConvLSTM2D with a (3,3) kernel are also an option for a more complex temporal encoder in those cases. Training and testing were done on 64×64 crops extracted from the hand-delineated frames used for the detection task. Training crops and testing crops came respectively from training and testing time-lapses. Thus, there was 4053 time-lapses (Fig 5) for training and 1908 time-lapses for testing.

For this multi-class problem, parse categorical cross entropy was used as the loss function with an Adam optimizer [51] and a reduce-on-plateau learning rate scheduler with an early stopping call-back. Each training lasted at most 30 epochs and stopped early if the validation

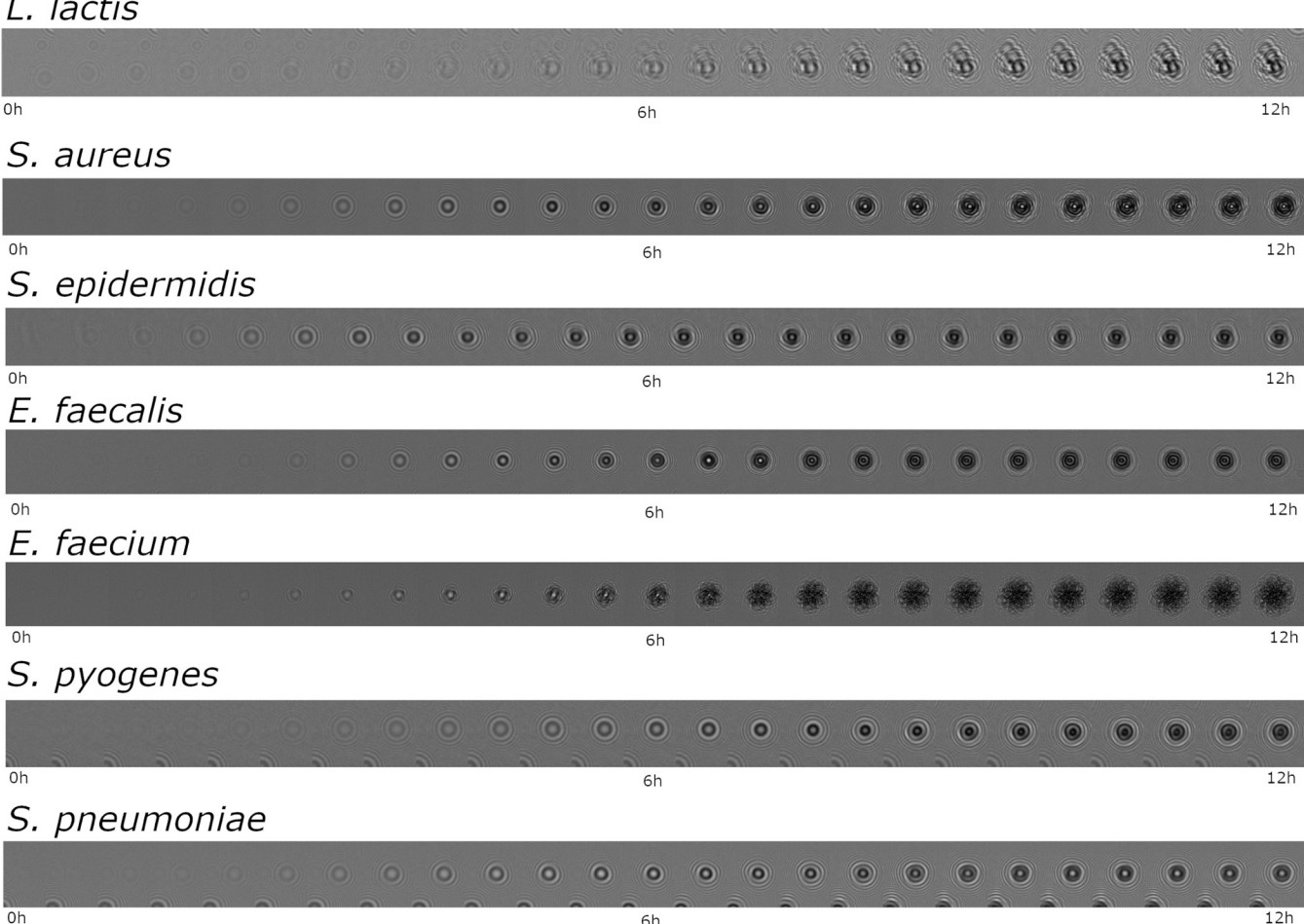

### L. lactis

0h 6h 12h

### S. aureus

0h 6h 12h

### S. epidermidis

0h 6h 12h

### E. faecalis

0h 6h 12h

### E. faecium

0h 6h 12h

### S. pyogenes

0h 6h 12h

### S. pneumoniae

0h 6h 12h

**Fig 5. Growth kinetics.** Growth kinetics of *L. lactis*, *S. aureus*, *S. epidermidis*, *E. faecalis*, *E. faecium*, *S. pneumoniae* and *S. pyogenes* over a twelve-hour incubation time with a 30-minute period between each frame. Each frame is a 64x64-pixel image (107x107 µm).

loss varied by less than 0.001 for 15 epochs. The starting learning rate (0.001) was multiplied by 0.2 whenever the validation loss reached a 10-epoch plateau.

Data augmentation was also employed to boost training samples numbers and to avoid overfitting with video augmentation techniques thanks to the *vidaug* python library [52]. This library lets us add random noise, move frames along the x-axis or y-axis and rotate images making up the time-lapses in a homogeneous manner.

## 5. Results and discussion

We analysed colonies from seven different species (*S. aureus*, *E. faecium*, *E. faecalis*, *S. epidermidis*, *S. pneumoniae*, *S. pyogenes*, *L. Lactis)* extracted from 13 experiments. Our testing dataset contained 350 images and 1908 colonies which were only used in this final evaluation of our networks' detection and classification performance. Due to bacterial concentration and experiment number, for each species there was a different amount of colonies to detect and classify. In fact, the classes in our test dataset are unbalanced in similar proportions to the training set: we have 275 colonies of S. *aureus* (2 experiments), 210 *E. faecium* (2 experiments), 60 *E. faecalis* (1 experiment), 647 *S. epidermidis* (2 experiments), 347 *S. pneumoniae* (1 experiment), 271 *S. pyogenes* (3 experiments) and 98 *L. lactis* colonies (2 experiments)—the remaining data being time-lapses of non-moving objects or empty areas.

To compensate for this class imbalance, our networks gave more importance to the classes with the least colonies during training. This way, trained weights will not favour one class over the others. However, it should be kept in mind that a class with more data will generally perform better than a class with less data and will be less sensitive to variations and errors.

We tested our networks performance on two tasks—detection and classification. The first task was evaluated by applying our Mask RCNN network on images at different time marks and comparing the bounding boxes it produced with our hand labelled ground truth. The second task was evaluated by carrying out inferences with our Conv3D+Lstm2D models on time-lapses varying in length but all beginning at the same time mark (t = 0h).

### 5.1 Detection

**5.1.1 Metrics.**   During the detection phase, the neural network task is to detect every growing colony. It is not a problem if some detected elements are not colonies but specks of dust as the next neural network will be able to differentiate between growing colonies and immobile structures. In other terms, high sensitivity (few false negatives) is the most important measure to maximise, even if it implies a low specificity (many false positives). By studying another criterion—the rate of correctly detected colonies over the total number of actual bacterial colonies present in the culture medium—the neural network effectiveness can be quantified and visualized simply with the data provided.

Another important output to measure is ROIs (regions of interest) prediction quality, for that matter the IOU criterion is used, which consists of calculating the ratio between actual and predicted areas. This score helps understanding how accurate correct ROIs are. It is normally considered that an IOU score under 0.3 is poor while scores over 0.7 are quite good [53]. However, the way each ground truth was created (by circling around full holographic 12h growth pattern, at their biggest) forces us to relativize the score obtained as IOU decrease very quickly. In this case, an IOU score above 0.5 can be considered as quite good while scores under 0.2 translate a really poor detection.

**5.1.2 Results.**   The trained detection network was tested upon a dataset consisting of 1902 colonies from seven different species. This dataset was tested at six different marks in time (2,

4, 6, 8, 10 and 12 hours) to compare and quantify its detection quality over time. True positives detection quality was also evaluated thanks to an IOU score.

The results presented in Fig 6 are from inferences utilising a specific version of the detection network. As explained earlier, its training is comprised only of frames between the 5th and 10th hour marks. Thus, decent results are expected from the 5th hour onwards; however, the detection rate reached an already quite decent quality at the 4th hour. At T = 2h (i.e. 5 frames) the network averaged at a 6.8% detection rate with low IOU scores (0.27 in average) indicating that these detections were mainly accidentals and uncorrelated to the presence of micro-colonies. Two hours later, at T = 4h, there is already an average 55.8% detection rate with decent IOU (0.42 in average), which is a huge rise in detection quality. We attain satisfactory detection rates from T = 8h onwards with an average detection at 96.2% and a good IOU score of 0.51, enough to have well-centred crops to extract for the next network.

The best average detection rate (96.9%) is reached after 12h of incubation with IOU scores ranging from 0.32 (*E. faecium*) to 0.66 (*S. pyogenes*) meaning that detections are in average quite accurate. From those observations, it can be seen that this network performs well during the 5-10h period for which it was designed. Moreover, it is already capable of proposing a reasonable detection, although incomplete, from the 4th hour of growth onwards while its best results are on data acquired at T = 12 h. This shows that our choice to train on a specific period of colony growth was a coherent decision. It focused the training on relevant data and helped generalize the representations in the neural network, which allowed for better and more consistent results earlier.

Those results seem coherent with what can be observed on Fig 5 time-lapses for instance. Species such as *S. epidermidis* or *S. aureus* produce colony holograms visible as early as early as 4h while some as *L. lactis* or *S. pyogenes* take longer times (~6/7h) to develop into a well-contrasted colony. That is why our performance began to rose around the 4th hour and soared at the 6th hour. Results continued to improve for slower strains up to 12 hours whereas fast growing bacteria reached and stabilized around 100% detection rate under 6 hours.

Theses detection results places us on par with other state-of-the-art rapid bacterial identification techniques, nearing Wang et al. [32] work proposing a full detection (~100% on every strain) in 6h hours, which is a very impressive achievement even if we take into consideration their restricted (only 3 *Enterobacterales*) and larger dataset (~16000 colonies to train on).

## 5.2 Classification

**5.2.1 Metrics.** During the classification phase, our neural network's goal is to correctly label every time-lapse used to produce inferences in a rapid fashion. This translates into maximising the number of true positives. True positives predictions can be evaluated by looking at sensitivity (or recall) and precision (or positive predictive value). Sensitivity, also called true positive rate (TPR), quantifies the proportion of true positive predictions. Precision, or positive predictive value (PPV), measures the rate of false positives. They respectively are defined by $TPR = \frac{TP}{TP+FN}$ and $PPV = \frac{TP}{TP+FP}$, where TP refers to the number of true positives, FP to the number of false positives and FN to the number of false negative predictions.

**5.2.2 Results.** Looking at the average precision and sensibility of our seven pathogenic bacteria over time (Fig 7), it can be established that the classification network steadily improves during incubation going from approximatively 15% sensitivity and 25% precision during the first hours (1–3 h) to an average 93.8% for sensitivity and 93.5% precision at 12h. However, precision and sensitivity trends are different for each species (S2 Fig and S3 Fig) and looking only at average results is not enough to understand the network outputs. For most species there is a sudden rise in sensitivity and precision between 4 and 7 hours and then a transition

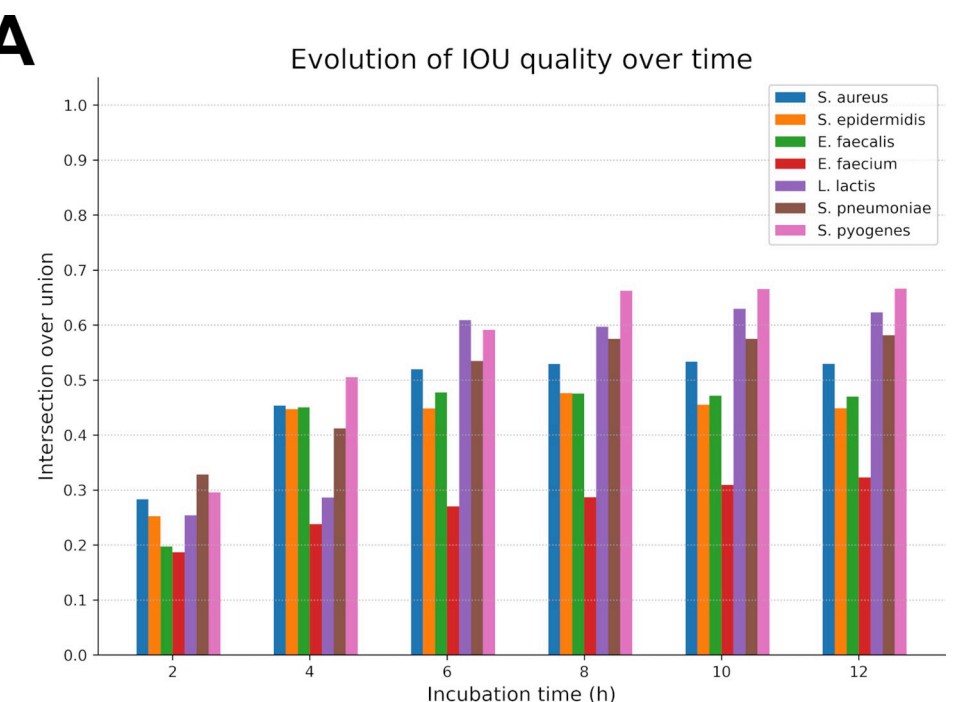

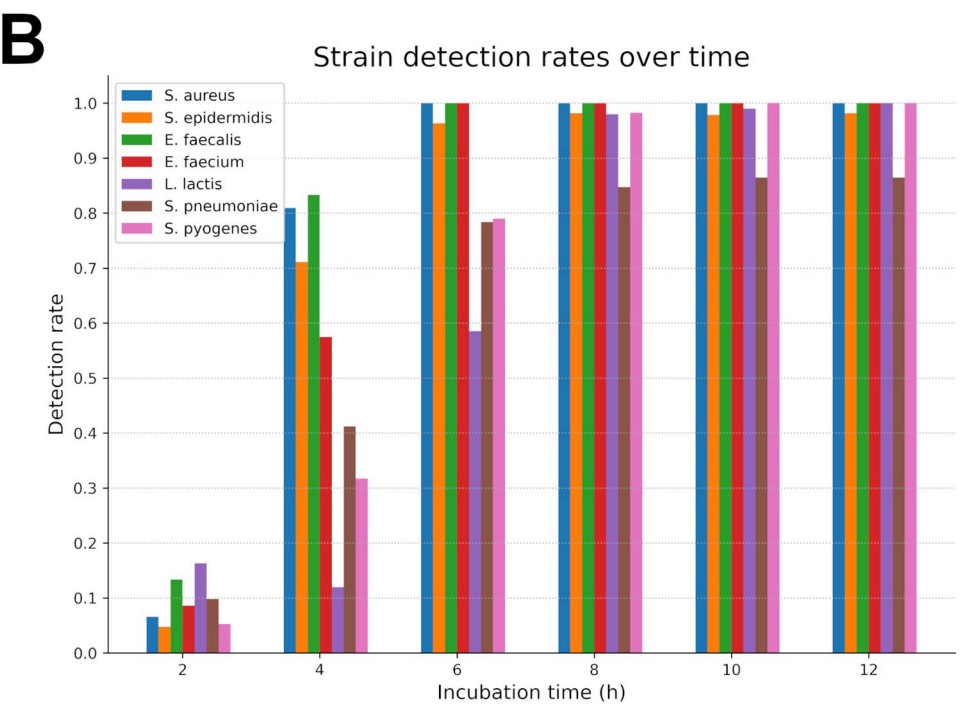

**Fig 6. Detection phase results.** (A) Detection rates and (B) IOU scores at four specific time marks (2, 4,6, 8,10, 12) for every strain.

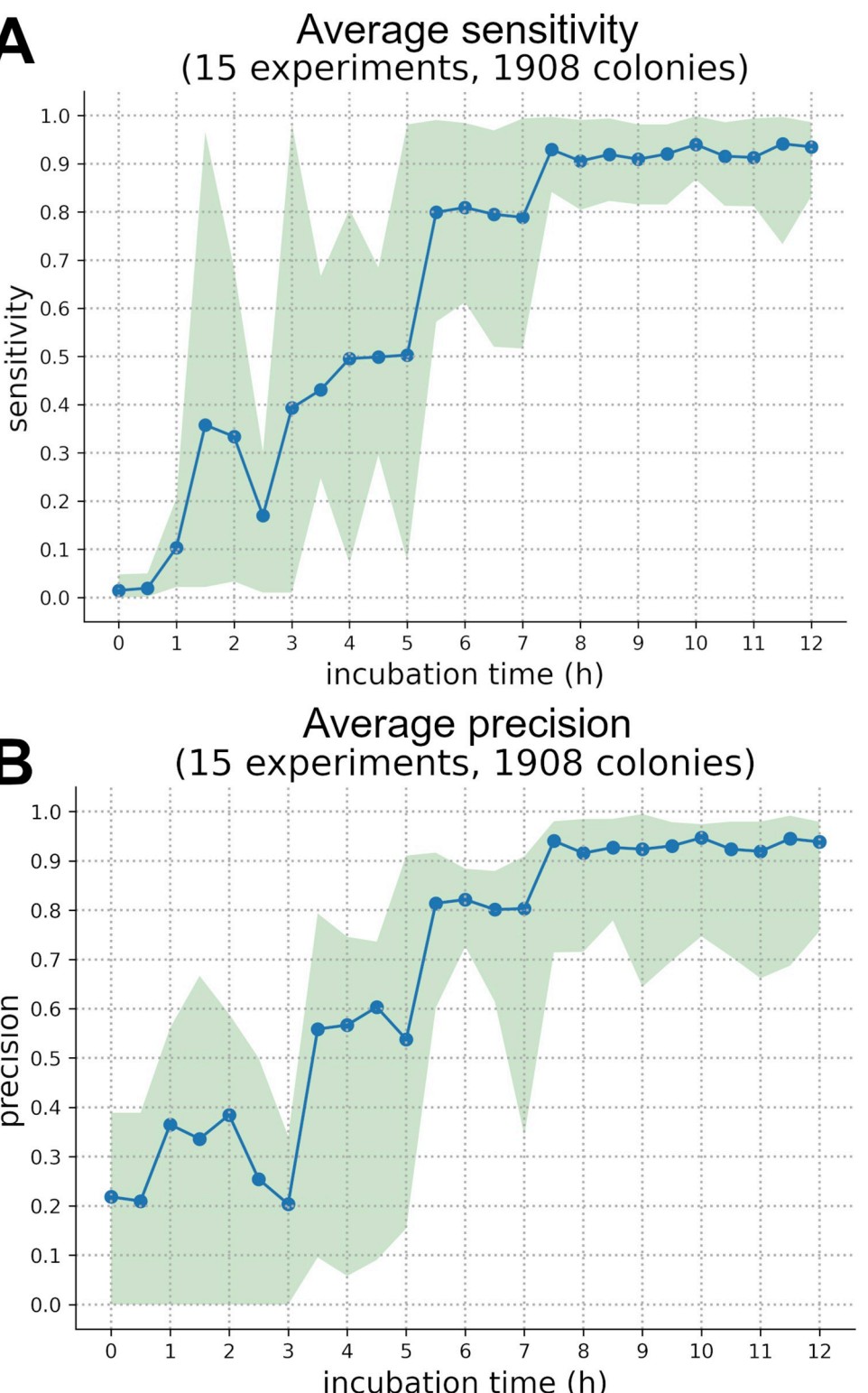

**Fig 7. Classification phase average precision and sensitivity.** Precision and sensitivity performance during the classification phase over a 12-hour incubation period on average. The green area represents the variation between the maximum and the minimum sensitivity for each time mark on all species.

to a stable plateau, meaning that the classification network correctly discriminates those species thanks to their growth kinetic and holographic pattern.

There is such a sudden rise in sensitivity for *S. pneumoniae*, *L. lactis*, *E. faecium*, *S. pyogenes* and *S. aureus* between 5h and 7h while for *E. faecalis* and *S. epidermidis* this increase happens earlier, respectively during 3-6h and 1-5h periods (S3 Fig).

For *E. faecium*, *E. faecalis*, *L. lactis* and S. *aureus* rises in precision and sensitivity are nearly correlated, only *E. faecalis* sees a synchronised rise in precision and sensitivity which might be due to its small number of colonies in the test dataset. Both *S. pneumoniae* and *S, pyogenes* have an earlier increase in their precision scores by 1 or 2 hours while S. Epidermidis sees its precision lagging behind its sensitivity by several hours. From this analysis, we can suppose that there was a S. epidermidis over-guessing in early models which is explainable by the fact that this species is the most numerous in our dataset (647 test colonies). From those trends it can be inferred that each strain starts to differentiate itself from the rest of the dataset at a specific time; *S. aureus*, *E. faecalis*, *E. faecium*, *S. epidermidis*, *S. pneumoniae*, *L lactis and S. pyogenes* respectively do so at 4h, 5h, 6h, 6h, 7h and 7h.

Those trends show that growth kinetics might be a suitable way to discriminate bacteria in an early fashion. However, an average 93.1% in precision and a 94.0% sensitivity is not enough to produce very high-quality inferences. Having said that, our results are on an equal footing with the rapid bacterial optical identification literature: Wang *et al.* [32] lens-free method achieves a 93.8% accuracy (3 species, 955 colonies) after a 12h incubation while Tang et al. [25] forward scattering technique using BARDOT reaches ~90% accuracy in less than 24h (37 strains ~3700 colonies). Nevertheless, our dataset still needs to be larger to compensate for data unbalance and to achieve better results.

Indeed, performance is very species-dependent and varies greatly with each strain; therefore–after a 12-hour incubation–precision can range from 83.3% for *E. faecalis* (60 colonies) to ~100% for *E. faecium* (99.%, over 210 colonies) and S. epidermidis (98.6% over 647 colonies), while sensitivity varies between 75.6% for *E. faecalis* and 96.7% for *S. Epidermidis*. This phenomenon is due to the data itself, its quality and quantity in particular; there is a clear correlation between number of colonies and the results performance. Species with more colonies to test on (which also means more to train on) such as *S. epidermidis* (647 colonies) and *S. aureus* (275 colonies) have better precision and sensitivity in those 12-hour tests while the hardest species to correctly identify was *E. faecalis* with much less data than other species.

*E. faecalis* was indeed much harder to correctly label at each time mark than other species but if confusion matrixes (Fig 8) are taken in account for the 8[th] and 12[th] hour time marks it can be understood that *E. faecalis* crops are not just randomly assigned to another label. In fact, there are two distinguishable parts in those matrices. There is a clear distinction between different genera and thus errors are mostly between similar genera: *E. faecalis* crops are primarily confused with *E. faecium* colonies. Similarly, confusions take place between staphylococci (S. epidermidis and S. aureus labels are mixed). This could mean that holographic patterns share similarities between genera and are not just species dependant; this would allow for a two-time detection with first the correct genus and then the right species. This would be done by using the possibilities of the COCO format by introducing super-categories to generalize between multiple classes and implementing a class hierarchy as done in W.Goo et al. paper [54] to learn discriminative subcategory features as different as possible from grouped super-category features.

## 6. Conclusion

Using live-cell lens-free imaging system in visible range (636 nm) and thin-layer (150μm) BHI agar we have acquired the very first database on anaerobic pathogenic bacteria from

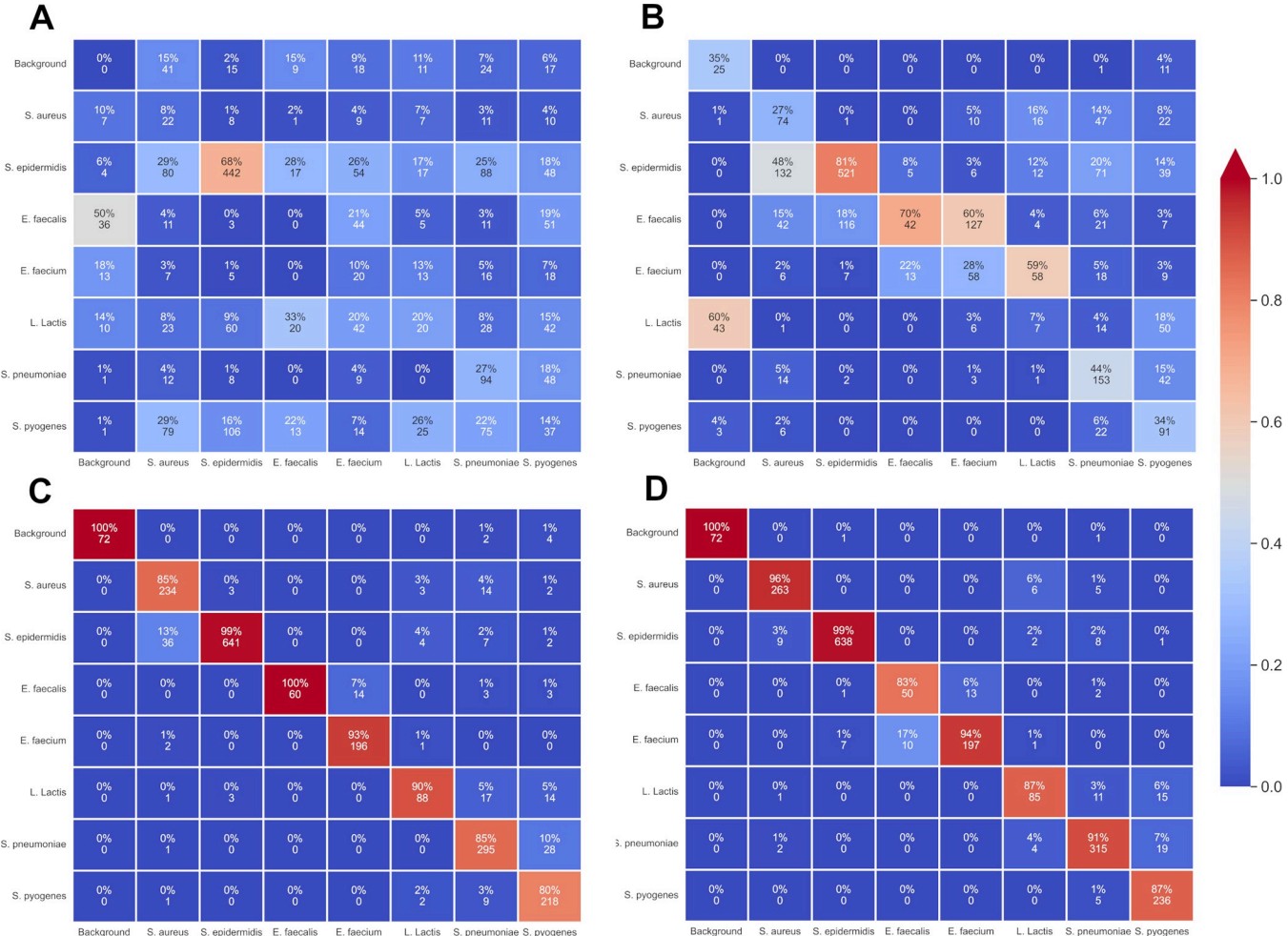

**Fig 8. Classification networks confusion matrices.** Classification networks confusion matrices at (A) T = 2h, (B) T = 4h, (C) T = 8h, (D) T = 12h on eight classes, which are Background, *S. aureus*, *S. epidermidis*, *E. faecium*, *E. faecalis L. lactis*, *S. pneumoniae* and *S. pyogenes*. Each matrix contains the predicted label of 275 S. *aureus*, 647 S. *epidermidis*, 60 *E. faecalis* and 210 *E. faecium*, 98 *L. lactis*, 347 *S. pneumoniae* and 271 *S. pyogenes*.

Enterococcus and Streptococcus genera. Our two-stage deep learning architecture (a mask convolutional network for detection and a three-dimensional recurrent network for identification) achieved an average 96.0% detection rate after an 8h incubation time (97.1% at 12h) and an average 93.6% correct identification rate with 93,1% precision and 94.0% sensitivity.

Those results were reached with holographic amplitudes and growth kinetic patterns only, proving un-reconstructed approach to be viable in certain setups for detection and identification purposes. Their features were extracted thanks to our novel proposal based on the coupling of recurrent and convolutional networks.

Thus, we were able to retrieve spatio-temporal features, which allowed for correct and rapid colony identification. Proving that holographic time-lapses combined with a deep learning pipeline to analyse those lens-free videos is an efficient way to answer the need for a solution that permits a rapid, high throughput, wide range, non-intrusive and non-destructive robust detection of every colony in a sample and satisfactory identification in real time.

Future investigations will both aim at analysing a wider surface and reaching better identification performance. New solutions are being developed to monitor 90-mm diameter Petri dishes thanks to in-house imaging sensors. A multi-spectral solution, which might provide

more phenotypic information, is also being considered in order to improve detection quality over a larger dataset.

## Supporting information

**S1 Table. Identification methods comparative table.** Pros and cons of techniques used for bacterial identification.
(TIFF)

**S1 Fig. Detailed detection and classification workflows.** Detailed workflow split in three main phases (Data preparation, Training and Testing) for both detection and classification tasks.
(TIFF)

**S2 Fig. Classification phase precision.** Precision performance during the classification phase over a 12-hour incubation period for (A) *E. faecium*, (B) *L. lactis*, (C) *E. faecalis*, (D) *S. Pyogenes* (E) *S. pneumoniae*, (F) *S. epidermidis* and on (G) *S. aureus*.
(TIFF)

**S3 Fig. Classification phase sensitivity.** Sensitivity performance during the classification phase over a 12-hour incubation period for (A) *E. faecium*, (B) *L. lactis*, (C) *E. faecalis*, (D) *S. Pyogenes* (E) *S. pneumoniae*, (F) *S. epidermidis* and on (G) *S. aureus*.
(TIFF)

## Author Contributions

**Conceptualization:** Paul Paquin, Caroline Paulus, Pierre R. Marcoux.

**Data curation:** Claire Durmort, Thierry Vernet.

**Formal analysis:** Paul Paquin.

**Funding acquisition:** Caroline Paulus, Pierre R. Marcoux, Sophie Morales.

**Methodology:** Paul Paquin, Claire Durmort.

**Project administration:** Sophie Morales.

**Resources:** Sophie Morales.

**Software:** Paul Paquin.

**Supervision:** Paul Paquin, Claire Durmort, Caroline Paulus, Pierre R. Marcoux, Sophie Morales.

**Validation:** Paul Paquin, Caroline Paulus, Thierry Vernet, Pierre R. Marcoux, Sophie Morales.

**Visualization:** Paul Paquin.

**Writing – original draft:** Paul Paquin.

**Writing – review & editing:** Paul Paquin, Caroline Paulus, Pierre R. Marcoux, Sophie Morales.

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
