## [Decision Letter · Decision Letter 0]

15 Mar 2022

PDIG-D-21-00104

Spatio-temporal based deep learning for rapid detection and identification of bacterial colonies through lens-free microscopy time-lapses

PLOS Digital Health

Dear Dr. PAQUIN,

Thank you for submitting your manuscript to PLOS Digital Health. After careful consideration, we feel that it has merit but does not fully meet PLOS Digital Health's publication criteria as it currently stands. Therefore, we invite you to submit a revised version of the manuscript that addresses the points raised during the review process.

We look forward to receiving your revised manuscript.

Kind regards,

Judy Wawira Gichoya

Section Editor

PLOS Digital Health

Journal Requirements:

1. Please update your Competing Interests statement. If you have no competing interests to declare, please state: “The authors have declared that no competing interests exist.”

Additional Editor Comments (if provided):

Thank you for the submission of your work 

We invite you to submit a revised manuscript to meet the points raised especially by reviewer #1

Reviewers' comments:

Reviewer's Responses to Questions

**Comments to the Author**

1. Does this manuscript meet PLOS Digital Health’s publication criteria? Is the manuscript technically sound, and do the data support the conclusions? The manuscript must describe methodologically and ethically rigorous research with conclusions that are appropriately drawn based on the data presented.

Reviewer #1: Partly

2. Has the statistical analysis been performed appropriately and rigorously?

Reviewer #1: No

Reviewer #2: Yes

3. Have the authors made all data underlying the findings in their manuscript fully available (please refer to the Data Availability Statement at the start of the manuscript PDF file)?

Reviewer #1: No

Reviewer #2: Yes

4. Is the manuscript presented in an intelligible fashion and written in standard English?

Reviewer #1: No

Reviewer #2: Yes

5. Review Comments to the Author

Reviewer #1: This work investigates about detection and identification of bacterial species directly looking at time-lapses of holographic signatures produced by lens-free microscopy during early stages of bacteria colony forming (up to 12 hours of growth). Despite the combination of this technique with deep learning-based analysis can be considered interesting the paper presents several issues that prevent considering this work mature enough for publication on PLOS Digital Health at the current stage. Following I indicate the main problems I found and I suggest the authors consider the possibility to revise (major revision) their work accordingly.

It would be interesting to have a better contextualization of your work based on a wider discussion (more than just the six lines (from 100 to106) about other optical (spatial-spectral) bacterial identification methods based on “colony fingerprints”, including for example 1) a wider discussion about forward light scattering research in the same application domain, also considering the contributions of other groups, like https://doi.org/10.1371/journal.pone.0105272 and https://doi.org/10.1016/j.bios.2015.01.047, and 2) other kind of spectral fingerprinting for the same identification aim obtained by hyperspectral imaging, as e.g. https://doi.org/10.1117/12.2306256 and https://doi.org/10.1016/j.compbiomed.2017.06.018 . In particular, despite the proposed lens-free approach seems particularly promising for early species identification, also the above methods has been proposed targeting at detection/identification of pathogens in earlier growing stages with respect to conventional reading times.

A more informative comparison of the various optical methods that can lead to bacterial identification could be a nice contribution. This could be based on a table with opportunities (pros) and limitations (cons) taking into account several desiderata, like turnaround time, automation, species discrimination power and related species and sub-species differentiation power, consumable constraints (agar type, growing media,…), …

The differentiation of the proposed method with respect to others already exploiting lens-free imaging is not clear since it is summarized in only 3 lines (from 117 to 119). This is not enough informative and not sufficient to assess the novelty of the work.

It is not clear whether the cultured bacteria come from ATCC strains or from clinical samples (the fact that the cultures are called “monomicrobial” could lead to think the first or a secondary culture after a first isolation). I think this is a relevant information and, in any cases, some discussion about possible changes in holographic patterns caused by serovars or subspecies should be given.

The dataset is limited in terms of number of pathogens and number of colonies. From this point of view this is only a preliminary work. The potential merit of the study to show a possible promising research path for rapid bacteria identification. However, the fact that authors decided to measure results on only 4 bacteria over 7 used for training is unconventional and objectionable. Results should include all 7 species with possible due considerations (as the ones given by the authors) about the fact that only 4 can be considered enough independent/reliable for testing, possibly proposing some results only for a subset related to truly independent datasets. I think that completely avoiding showing results for all the 7 species is not acceptable.

The description given from line 228 to 234 is unclear. A figure could help supporting reader understanding.

A figure describing the whole processing workflow lens-free image are subject to should be provided (what happens to images, how images are handled, cropped, sampled…. How images and subimages flow into ne networks) to support description given in sec. 4.5 and to add more info since fig.5 is not enough informative.

A general introduction to section 5 before starting with specialized sections is advisable.

What said from line 314 to 317 is not described in the methods. Is performance step up described at lines 340-343 due to this delta-growing verification mechanism? Is it possible consolidate early detection performance (T=4h) looking back from the future times. Moreover, I’m not sure to understand why, since you are interested to the grow dynamic (time-lapse) of the same colony in a certain time interval, you do not follow back the same colony starting from the max time (when detection performance are better) toward the min time. The same concerns apply to why results are divided in time steps in sec. 5.2.2. and fig.8,9,10. It is not reasonable that the same colony over time changed species, therefore I would only expect a unique consensus result for each colony.

Claim at lines 410-413 about the possibility of a two-step identification (first genus and then species) is not supported by methodological specifications. Please specify how this two-step decision is or could be made.

Overall, I do not find the use of deep learning solutions highly justified against more traditional image analysis or machine learning approaches. A deeper discussion on this is necessary along with clear indication about how authors prevent overfitting.

A thoughtful discussion of results (in a dedicated section again taking into account the literature) is lacking and should be added.

Quality (resolution) of figures is low and this impairs understanding especially of fig. 4 and 10.

Linguistic revision is needed since not always good/correct expressions are selected. Moreover, there are too many typos, then a careful proofreading is necessary. Among others: line 240: multiples predictions, line 260: border box � bounding box, performances�performance (in mos cases).

Reviewer #2: This research work is quite impressive, based on the results and the approach taken to attain such good output. There is good hope that this work can be extended to even achieve better identification and detection rates. However, there are areas that need some improvement/clarification as indicated below.

Please include the missing (critical) information in the abstract summary

Provide a justification on why you choose to use holographic non-reconstruction

Provide a justification for choosing Microsoft COCO format and not others

Correct the grammatical errors

Critically clarify on the issue of using a classification network that is dynamic with respect to holographic non-reconstruction. 

Provide a comparison table with other researchers approaches including their time taken to identify, the detection rates, and possibly which kind of data sets they were dealing with (size is important, because its difficult to compare with work which has bigger data sets with small data sets; definitely there is a big difference in results.

6. PLOS authors have the option to publish the peer review history of their article (what does this mean?). If published, this will include your full peer review and any attached files.

**Do you want your identity to be public for this peer review?** For information about this choice, including consent withdrawal, please see our Privacy Policy.

Reviewer #1: No

Reviewer #2: No

**Comments to the Author**

1. Does this manuscript meet PLOS Digital Health’s publication criteria? Is the manuscript technically sound, and do the data support the conclusions? The manuscript must describe methodologically and ethically rigorous research with conclusions that are appropriately drawn based on the data presented.

Reviewer #2: Yes

---

## [Decision Letter · Decision Letter 1]

3 Aug 2022

PDIG-D-21-00104R1

Spatio-temporal based deep learning for rapid detection and identification of bacterial colonies through lens-free microscopy time-lapses

PLOS Digital Health

Dear Dr. PAQUIN,

Thank you for submitting your manuscript to PLOS Digital Health. After careful consideration, we feel that it has merit but does not fully meet PLOS Digital Health's publication criteria as it currently stands. Therefore, we invite you to submit a revised version of the manuscript that addresses the points raised during the review process.

Please add more details to the GitHub repository README file to make it easy to navigate for those interested in using the files, and add all data used for the manuscript if possible. After these tasks are completed the manuscript will be accepted for publication.

Please submit your revised manuscript within 30 days Oct 02 2022 11:59PM. If you will need more time than this to complete your revisions, please reply to this message or contact the journal office at digitalhealth@plos.org. Please include the following items when submitting your revised manuscript:

We look forward to receiving your revised manuscript.

Kind regards,

Heather Mattie

Academic Editor

PLOS Digital Health

Journal Requirements:

Additional Editor Comments (if provided):

Revision has adequately addressed the concerns of the reviewers, but not all data has been made available and the GitHub repository provided that contains sample data needs a better README file - at the moment it is difficult to navigate. Please add more details to the repository README file and add all data used for the manuscript. After these tasks are completed the manuscript will be accepted for publication.

Reviewers' comments:

Reviewer's Responses to Questions

**Comments to the Author**

1. If the authors have adequately addressed your comments raised in a previous round of review and you feel that this manuscript is now acceptable for publication, you may indicate that here to bypass the “Comments to the Author” section, enter your conflict of interest statement in the “Confidential to Editor” section, and submit your "Accept" recommendation.

Reviewer #1: All comments have been addressed

Reviewer #2: All comments have been addressed

2. Does this manuscript meet PLOS Digital Health’s publication criteria? Is the manuscript technically sound, and do the data support the conclusions? The manuscript must describe methodologically and ethically rigorous research with conclusions that are appropriately drawn based on the data presented.

Reviewer #1: Yes

Reviewer #2: Yes

3. Has the statistical analysis been performed appropriately and rigorously?

Reviewer #1: Yes

Reviewer #2: N/A

4. Have the authors made all data underlying the findings in their manuscript fully available (please refer to the Data Availability Statement at the start of the manuscript PDF file)?

Reviewer #1: No

Reviewer #2: Yes

5. Is the manuscript presented in an intelligible fashion and written in standard English?

Reviewer #1: Yes

Reviewer #2: Yes

6. Review Comments to the Author

Reviewer #1: The paper has been significantly improved and I thank the authors for having seriously considered and taken actions on my previous concerns. I suggest further proofreading since I found few typo. E.g. line 411 what can be observe  what can be observed.

What refrain me to suggest immediate acceptance is the fact that Plos Digital Health policy about data availability require the full set of data necessary to replicate the study findings to be available, while authors uploaded only a portion of their dataset and no network models has been shared. Therefore replicability is not guaranteed at the moment.

Reviewer #2: The Authors have been able to respond to comments, i gave.

Looking through also the comments given by the other reviewer, to me, these i see have also been addressed

Generally, the article is now in a good shape to be accepted. This is preliminary and really paves a good starting point for other researchers to use more data and build on and come up deeper thoughts and solutions in the same area.

7. PLOS authors have the option to publish the peer review history of their article (what does this mean?). If published, this will include your full peer review and any attached files.

**Do you want your identity to be public for this peer review?** For information about this choice, including consent withdrawal, please see our Privacy Policy.

Reviewer #1: No

Reviewer #2: No

---

## [Editor Report · Decision Letter 2]

9 Sep 2022

Spatio-temporal based deep learning for rapid detection and identification of bacterial colonies through lens-free microscopy time-lapses

PDIG-D-21-00104R2

Dear Mr. PAQUIN,

We are pleased to inform you that your manuscript 'Spatio-temporal based deep learning for rapid detection and identification of bacterial colonies through lens-free microscopy time-lapses' has been provisionally accepted for publication in PLOS Digital Health.

Best regards,

Heather Mattie

Academic Editor

PLOS Digital Health